# Insights into the Novel *FAD2* Gene Regulating Oleic Acid Accumulation in Peanut Seeds with Different Maturity

**DOI:** 10.3390/genes13112076

**Published:** 2022-11-09

**Authors:** Shuzhen Zhao, Jie Sun, Jinbo Sun, Xiaoqian Zhang, Chuanzhi Zhao, Jiaowen Pan, Lei Hou, Ruizheng Tian, Xingjun Wang

**Affiliations:** 1Institute of Crop Germplasm Resources, Shandong Academy of Agricultural Sciences, Shandong Provincial Key Laboratory of Crop Genetic Improvement, Ecology and Physiology, Jinan 250100, China; 2College of Life Sciences, Shandong Normal University, Jinan 250014, China

**Keywords:** peanut, FAD2, seed maturity, oleic acid

## Abstract

AhFAD2 is a key enzyme catalyzing the conversion of oleic acid into linoleic acid. The high oleic acid characteristic of peanut mainly comes from the homozygous recessive mutation of *AhFAD2A* and *AhFAD2B* genes (aabb). However, even in high-oleic-acid varieties with the aabb genotype, the oleic acid content of seeds with different maturity varies significantly. Therefore, in addition to *AhFAD2A* and *AhFAD2B*, other *FAD2* members or regulators may be involved in this process. Which *FAD2* genes are involved in the regulatory processes associated with seed maturity is still unclear. In this study, four stable lines with different genotypes (AABB, aaBB, AAbb, and aabb) were used to analyze the contents of oleic acid and linoleic acid at different stages of seed development in peanut. Three new *AhFAD2* genes (*AhFAD2–7*, *AhFAD2–8*, and *AhFAD2–9*) were cloned based on the whole-genome sequencing results of cultivated peanuts. All peanut *FAD2* genes showed tissue preference in expression; however, only the expression level of *AhFAD2-7* was positively correlated with the linoleic acid concentration in peanut seeds. These findings provide new insights into the regulation of oleic acid accumulation by maturity, and AhFAD2-7 plays an important role in the maturity dependent accumulation of oleic acid and linoleic acid in peanut.

## 1. Introduction

Peanut (*Arachis hypogaea* L.) is one of the most important oil crops worldwide. Oleic acid (18:1) and linoleic acid (18:2) account for more than 80% of the total fatty acids in peanut. The oxidation stability of oleic acid is 10 times higher than that of linoleic acid; therefore, peanuts with high oleic acid and its products have a longer shelf life [1]. In addition, compared with normal peanuts, high-oleic-acid peanuts and their products are beneficial to the health of the cardiovascular system [2]. FAD2, Δ12—fatty acid desaturase, a key enzyme, catalyzes the conversion of oleic acid to linoleic acid and determines the fatty acid composition in plants. Two types of Δ12—fatty acid desaturase were identified in plants based on their subcellular localization and electron donor [3]. One type is the microsomal Δ12—fatty acid desaturase (FAD2), located in the endoplasmic reticulum, with cytochrome b5 as the electron donor. The other type is the plastid Δ12—fatty acid desaturase (FAD6), located in the plastids, with ferridoxin as the electron donor [4]. Previous studies have shown that the gene expressions of some microsomal fatty acid dehydrogenases are seed specific and others are constitutive [5,6].

Since the first *FAD2* gene cloned from *Arabidopsis*, *FAD2* genes have been successively identified and characterized in different plant species. Only one copy of the *FAD2* gene was identified in *Arabidopsis*, expressed in all tissues and involved in the synthesis of both seed storage lipids and membrane lipids. The transcriptional expression of the *AtFAD2* gene was not affected by low temperature [7]. A varied number of *FAD2* genes were identified in soybean, sunflower, cotton, and other crops. Four *FAD2* genes were cloned in soybean, including two seed-specific genes, *FAD2-1A* and *FAD2-1B*, which were mainly responsible for the transformation of oleic acid to linoleic acid in seeds. The other two, *FAD2-2* and *FAD2-3,* were constitutively expressed in different tissues [8,9]. Simultaneous mutations of *FAD2-1A* and *FAD2-1B* by transcription activator-like effector nucleases (TALENs) could significantly increase the oleic acid content of soybean [10]. Three *FAD2* genes were identified in sunflower, among which *FAD2-1* was involved in the synthesis of seed-specific fatty acids and *FAD2-2* and *FAD2-3* were constitutively expressed [5]. Complementary experiments showed that the grape *FAD2* gene could restore the oleic acid content to its normal level in the *Arabidopsis FAD2* mutant [11]. Upland cotton contains four *FAD2* genes (*GhFAD2-1* to *GhFAD2-4*). *GhFAD2-1* was highly expressed in seeds as a seed-specific fatty acid dehydrogenase. *GhFAD2-3* was expressed in all analyzed tissues, such as roots, leaves, flower buds, fibers, and especially in the anther. Specific inhibition of *GhFAD2-3* by RNA interference led to male sterility and impaired anther development from meiosis to maturity [12,13]. Five *FAD2* genes were identified in olive (*OeFAD2-1* to *OeFAD2-5*). *OeFAD2-1* was mainly involved in the synthesis of seed oil, and *FAD2-2* was mainly expressed in seeds, leaves, and the mesocarp at a mature stage [14]. *OeFAD2-3*, *OeFAD2-4*, and *OeFAD2-5* were mainly expressed in olive fruits. *OeFAD2-5* and *OeFAD2-2* played an important role in the synthesis of linoleic acid in the mesocarp and therefore in olive oil. The expression levels of *OeFAD2-3*, *OeFAD2-4*, and *OeFAD2-5* genes in olive fruit were not only regulated in different tissues and developmental stages but also regulated by environmental factors [15]. Four *FAD2* genes were cloned in rapeseed (*BnaFAD2.A5*, *BnaFAD2.A1*, *BnaFAD2.C5*, and *BnaFAD2.C1*). *BnaFAD2.A1* mutation by CRISPR/Cas9 resulted in a significant increase in the oleic acid content [16].

In peanut, *FAD2A* and *FAD2B* genes from subgenomes A and B, respectively, were cloned using *Arabidopsis FAD2* cDNA as a probe [17]. *FAD2A* substitutions of a single base (G to A) at 448 bp resulted in significantly reduced enzyme activity [18]. One base insertion at 442 bp of the *FAD2B* gene led to a frame shift and eventually caused the loss of FAD2B protein function. Simultaneous mutations of these two genes produced high-oleic-acid mutants [19,20,21]. Based on these mutations, four genotypes (wild-type (AABB), a single mutation on *FAD2A* (aaBB), a single mutation on *FAD2B* (AAbb), and a double mutation (aabb)) were obtained in this study.

Significant differences were found in oleic acid/linoleic acid (O/L) in different peanut varieties, including high-oleic-acid varieties (aabb), and different maturity of seeds in the same variety. In addition to *AhFAD2A* and *AhFAD2B*, other *FAD2* genes or regulators could be involved in this process. Six members of the *FAD2* family were cloned from peanut and named *FAD2-1* to *FAD2-6* [22,23]. *FAD2-1*, *FAD2-2*, *FAD2-3*, and *FAD2-4* encode the endoplasmic reticulum omega-3 fatty acid dehydrogenase, while *FAD2-5* and *FAD2-6* encode the chloroplast omega-6 fatty acid dehydrogenase. Sequence comparison showed that *AhFAD2-1* is 76% homologous to *AhFAD2-2. FAD2-1* was mostly expressed in young seeds, while *FAD2-2* was mostly expressed in flowers [22]. However, which *FAD2* genes in peanut are involved in the transformation of oleic acid to linoleic acid in seeds related to maturity is still not clear. In this study, four stable *FAD2A* and *FAD2B* genotypes (AABB, aaBB, AAbb, and aabb) were used to analyze the changes in the oleic acid and linoleic acid contents in seeds at different developmental stages. After cloning the full length of peanut *FAD2* genes, the sequences of these genes were analyzed. In addition, the expression patterns of *FAD2* genes in these four genotypes and seeds at different developmental stages were investigated.

## 2. Materials and Methods

### 2.1. Plant Materials

Huayu 31 is a peanut variety with normal oleic acid content, while Kainong 176 is a peanut variety with high oleic acid content. Four *FAD2* genotypes (AABB, aaBB, AAbb, and aabb), obtained from BC_4_F_2_ lines constructed from the cross of Huayu 31 and Kainong 176, were used in this study.

### 2.2. Multiple Sequence Alignment and Phylogenetic Analysis

Clustalw was used to performed sequence alignments. CDS sequences of peanut *FAD2* family genes were searched from the peanut genome website (https://www.peanutbase.org (accessed on 17 November 2017)), and the protein domains of peanut FAD2 candidates were analyzed using SMART software (http://smart.embl-heidelberg.de/ (accessed on 10 October 2018)). The protein sequences containing PLN02505 and Δ12-FADs-like domains were considered as the protein sequences of peanut FAD2. *FAD2* sequences of *Arabidopsis thaliana*, *Arachis duranensis*, *Arachis ipaensis*, *Brassica rapa*, *Brassica napus*, *Glycine max*, *Gossypium hirsutum*, *Helianthus annuus*, *Ricinus communis*, *Ricinus communis*, *Sesamum indicum*, and *Solanum commersonii* were searched in the National Center for Biotechnology Information (NCBI) database (https://www.ncbi.nlm.nih.gov/ (accessed on 5 July 2021)). The conserved domains of the putative FAD2 desaturase enzymes were searched in the SMART databases (http://smart.embl-heidelberg.de/ (accessed on 15 December 2018)). Phylogenetic trees based on the alignment of conserved domain sequences were constructed using the neighbor-joining (NJ) algorithm with the bootstrap set to 1000 and the rest of the parameters at default settings in MEGA 6.0 [24].

### 2.3. Total RNA Extraction and cDNA Synthesis

Total RNA from peanut seeds at four development stages was extracted using the Aidlab RNA extraction kit (Takara Bio Inc., Beijing, China) and treated with DNase I to remove the contaminated genomic DNA according to the manufacturer’s protocols. The first-strand cDNA was synthesized for each sample using the Primer-ScriptTM 1st Strand cDNA Synthesis Kit (Takara Bio Inc., Beijing, China) and the oligo (dT) primer.

### 2.4. PCR Amplification and Sequence Analysis of Novel FAD2 Genes

To confirm novel *FAD2* genes of peanut, putative *FAD2* cDNA fragments were amplified by PCR using the cDNAs of the peanut genotype with AABB and aabb as templates. For this PCR, the primers were designed based on the cDNA of *AhFAD2-7*: *Arahy.BY45PL*, *AhFAD2-8: Arahy.S1Y1PZ*, and *AhFAD2-9*: *Arahy.9P5B67* available in the PeanutBase (Appendix A). PCR products were sequenced, and the sequences were BLASTed using the NCBI database to confirm their identity.

### 2.5. Real-Time PCR

Primer 5.0 software was used to design gene-specific primers for qRT-PCR analysis. The primer sequences are listed in Appendix A. The reaction mixture consisted of 2 μL of 50-fold diluted first-strand cDNA, 0.5 μL of 10 μM of each primer, and 10 μL of 2× FastStart Universal SYBR Green Master Mix (Roche, Indianapolis, IN, USA). The following PCRs were performed using An ABI 7500 real-time PCR system (Thermo Fisher Scientific, Waltham, MA, USA), and the program was 95 °C for 10 min, then 40 cycles of 95 °C for 15 s, and 60 °C for 1 min. Peanut *actin* was used as the reference gene for normalization. Non-specific products were identified by melting curve analysis. The 2^−ΔΔCT^ method was used for the calculation of relative expression levels [25].

### 2.6. Arabidopsis Transformation

Using the improved *Arabidopsis* floral dipping method described by Clough et al. [26]. Briefly, 2 mL of activated GV3101 containing the expression vector plasmid was added to 200 mL of YEP culture medium containing 50 µg/mL of kanamycin and 50 µg/mL of rifampin. The agrobacteria were cultured at 28 °C to an OD600 value of 1.0–1.2. The cultures were collected by centrifugation and then suspended with the infection solution (5% sucrose, 0.02–0.03% Silwet L-77) to a final OD600 value of 0.8. Inflorescences were immersed in the infection solution for 30 s and then cultured in the dark for 24 h before being transferred to a growth chamber.

### 2.7. Vector Construction and the Expression of the FAD2 Gene in S. cerevisiae

*Saccharomyces cerevisiae* INVSc-1 (Invitrogen, Waltham, CA, USA) was used for the expression of AhFAD2 and production of linoleic acid. Yeast transformation was carried out in accordance with the manufacturer’s manual, and yeast culture and induction of AhFAD2 were performed following Covello and Reed [27]. The specific primers containing KpnI and Xbal digestion sites were used for the PCR amplification of *AhFAD2-7*, *AhFAD2-8*, and *AhFAD2-9* coding regions, separately. The amplified *AhFAD2-7*, *AhFAD2-8*, and *AhFAD2-9* genes were digested using KpnI and Xbal and connected to the plasmid pYES2 dual-digested with the same restriction enzymes. Using the lithium acetate method, the constructs pYFAD2-7, pYFAD2-8, and pYFAD2-9, as well as the control vector pYES2, were transformed into INVSc1 cells [28]. Transformants were selected on minimal medium plates lacking uracil (SC-Ura). The yeast cells at the logarithmic phase were incubated at 25 °C for 48 h. The contents of oleic acid and linoleic acid in yeast were analyzed using gas chromatography.

## 3. Results

### 3.1. Fatty Acid Content in Peanut Seeds of Four FAD2 Genotypes at Different Development Stages

In this study, peanut pod development was divided into four stages according to the color and thickness of the mesocarp, namely white and thick mesocarp stage (S1), white and thin mesocarp stage (S2), brown and thin mesocarp stage (S3), and black and thin mesocarp stage (S4); see Figure 1. Fatty acids were determined using gas chromatography. Oleic acid, linoleic acid, and palmitic acid accounted for 90% of the fatty acids in peanut, and changes in these three major components were analyzed in this study.

The results showed that the genotypes of *AhFAD2A* and *AhFAD2B* significantly affect the contents of oleic acid, linoleic acid, and palmitic acid. At the same stage of pod development, the oleic acid content of the wild-type AABB was the lowest, while that of the homozygous mutant aabb was the highest. The oleic acid content of single mutations of *AhFAD2A* and *AhFAD2B* also significantly increased (Figure 2). Regardless of genotype, seed maturity significantly affected the contents of oleic acidt, linoleic acid, and palmitic acid. The oleic acid content and the ratio of the oleic acid content to linoleic acid (O/L) increased with seed maturity. The results indicated that other members of *AhFAD2* or other regulators, in addition to *AhFAD2A* and *AhFAD2B*, may be involved in the synthesis of linoleic acid during the maturation of peanut seeds.

### 3.2. Cloning of New Members of the FAD2 Gene Family

Three new *AhFAD2* gene members (*Arahy.BY45PL*, *Arahy.S1Y1PZ*, and *Arahy.9P5B67*) were identified through whole-genome sequencing BLAST using *AhFAD2A* CDS. ProtComp software predicted that the proteins encoded by these three genes are located on the endoplasmic reticulum membrane. These three genes, renamed as *AhFAD2-7*, *AhFAD2-8*, and *AhFAD2-9*, respectively, were cloned by the amplification of cDNA using gene-specific primers derived from the *Arahy.BY45PL*, *Arahy.S1Y1PZ*, and *Arahy.9P5B67* sequences. The length of *AhFAD2-7*, *AhFAD2-8*, and *AhFAD2-9* was 1152 bp, 1164 bp, and 1164 bp, respectively. The amino acid sequences of these genes were deduced. As a result, the lengths of these proteins were 383 aa, 387 aa, and 387 aa, respectively. The corresponding molecular weights were 43.8 kDa, 45.1 kDa, and 45 kDa, and *pI* values were 8.8, 9.04, and 8.96, respectively. The *AhFAD2-7* gene was located on chromosome B06, while *AhFAD2-8* and *AhFAD2-9* were located on chromosome A09 and B09, respectively. The alignment of the peanut FAD2 amino acid sequences deduced (Figure 3) showed that AhFAD2-7 displays 94.54% identity to the previously described AhFAD2-1 and shares 71.46% and 71.22% identity with AhFAD2A and AhFAD2B, respectively. Among them, AhFAD2A/B and FAD2-7 shared 64.02% and 68.98% identity with AhFAD2-8, respectively, while AhFAD2-8 and AhFAD2-9 shared 95.29% identity. The amino acid sequences of all AhFAD2 and AtFAD2 proteins were compared, and the result showed that all the predicted H-box regions are almost identical. Among AhFAD2 and AtFAD2 proteins, both the positions and numbers of H boxes and histidines in each H box were completely identical. In peanut, the sequences of the first two H boxes (closest to the N-terminus) of AhFAD2 isozymes are HECGHH and HRRHH, which are conserved and the same as with AtFAD2. Intriguingly, the third motif of AhFAD2-8 and AhFAD2-9 is HVTHH (closest to the C-terminus), different from other AhFAD2s and AtFAD2 with HVAHH. Because FAD2 is also an ER-localized fatty acid desaturase, the ER retrieval motifs in AhFAD2A, AhFAD2B, and AhFAD2-3 are all YKNKF. The ER retrieval motifs in AtFAD2, AhFAD2-1, and AhFAD2-7 are YNNKL, while those in AhFAD2-8 and AhFAD2-9 are YNKL.

### 3.3. Phylogenetic Relationship of FAD2 Genes

To elucidate the phylogenetic relationship between members of the FAD2 family in peanut and other species, phylogenetic analysis of the predicted amino acid sequences of the nine peanut genes and other 22 FAD2s from higher plants was performed using MEGA 6.05 and inferred using the neighbor-joining tree method with a bootstrap value of 1000. Phylogenetic analysis showed that peanut FAD2 members are divided into two distinct clades. The first clade comprised FAD2 enzymes, such as AhFAD2A, AhFAD2B, AhFAD2-3, AiFAD2, and AdFAD2, exhibiting a similar pattern of seed-specific expression. Similarly, FAD2 enzymes from soybean (GmFAD2-1A and GmFAD2-1B), cotton (GhFAD2-1), sesame (SiFAD2), and sunflower (HaFAD2-1) were all grouped together in this clade, which were also reported to be expressed only in the seeds (Figure 4). These seed-specific FAD2 genes were most likely playing major roles in converting oleic acid to linoleic acid during storage lipid biosynthesis in peanut seeds, while AhFAD2-1, AhFAD2-7, AhFAD2-8, and AhFAD2-9 were predicted to express constitutively in peanut.

### 3.4. Functional Verification of FAD2 Members in Saccharomyces cerevisiae and Arabidopsis

There is ample oleic acid that can serve as a precursor for FAD2 enzymes but no fatty acid dehydrogenase in *Saccharomyces cerevisiae* INVSc1. Heterologous expression in INVSc1 was used to confirm the functions of AhFAD2-7, AhFAD2-8, and AhFAD2-9 genes. The linoleic acid content accounted for 10.44%, 49.14%, and 57.03% of total fatty acids in the transformed yeast expressing AhFAD2-7, AhFAD2-8, and AhFAD2-9. Almost no linoleic acid was detected in the transformant containing only the empty vector (Figure 5, Table 1). These results demonstrated that FAD2-7, FAD2-8, and FAD2-9 have the catalytic activity of fatty acid dehydrogenase.

The *Arabidopsis fad2* mutant exhibits low linoleic acid content due to a point mutation in the FAD2 gene. The open reading frames (ORFs) of AhFAD2-7, AhFAD2-8, and AhFAD2-9 were transformed to the *Arabidopsis fad2* mutant. The linoleic acid content was 7.49%, 3.82%, and 13.06% in the seeds of three transgenic lines containing AhFAD2-7, AhFAD2-8, and AhFAD2-9 ORF, respectively (Appendix A, Table 2). These results indicated that peanut FAD2-7, FAD2-8, and FAD2-9 can indeed catalyze the conversion of oleic acid to linoleic acid in the *Arabidopsis fad2* mutant.

### 3.5. Expression Analysis of AhFAD2 Genes in Different Tissues and Seeds of Different Maturity

To evaluate the expressions of these *AhFAD2* genes, qRT-PCR analysis was carried out with total RNA extracted from the root, leaf, flower, and seeds of different maturity in four *FAD2* genotypes. *AhFAD2A* and *AhFAD2B* are highly homologous, with only 11 base differences. The designed primers detect the expressions of both *AhFAD2A* and *AhFAD2B* simultaneously. *AhFAD2A/B* and *AhFAD2-3* were specifically expressed in developing seeds, while their expression in root and leaf tissues was hardly detected (Figure 6). *AhFAD2A/B* and *AhFAD2-3* were not expressed in the stem. *AhFAD2-7* showed high expression in young seedling tissues, including the root, stem, and leaf, as well as the flower. *AhFAD2-7* expression was also observed in developing seeds, especially in the early developmental stages. *AhFAD2-8* and *AhFAD2-9* were mostly expressed in the flower, with relatively low levels in other tissues of all the tested *FAD2* genotypes.

The expression of *AhFAD2A/B*, *AhFAD2-3*, and *AhFAD2-7* was detected in all four developmental stages (S1–S4) of seeds. With the development of the seed, the expression trend of *AhFAD2A/B* and *AhFAD2-3* first increased and then decreased. On the contrary, the expression trend of *AhFAD2-7* decreased with maturity. *AhFAD2-8* showed almost no expression in S1 seeds. Low yet detectable levels of *AhFAD2-8* expression were observed in developing seeds, especially in the late developmental stages. We found that the expression level of *AhFAD2-7* is positively correlated with the oleic acid content in peanut seeds at different developmental stages (Figure 2).

## 4. Discussion

Several factors, such as the genotype, growing conditions, and maturity, affect the fatty acid content in peanut oil [29,30,31]. In this study, it was observed that the oleic acid content in different *AhFAD2* genotypes is significantly different during all developmental stages of the seeds. As expected, peanuts with the genotype aabb possessing both the mutant alleles of *ahFAD2A* and *ahFAD2B* produced the highest amount of oleic acid compared to AABB, aaBB, and AAbb. Similar results were also observed in FAD2 double-mutant soybean [32]. Single-mutation *AhFAD2A* (aaBB) or *AhFAD2B* (AAbb) also increased the oleic acid content. However, the contribution of the double mutant was much greater than the cumulative individual effect of FAD2 genes [33]. A previous study estimated that *ahFAD2B* contributes more to oleic acid accumulation compared to *ahFAD2A* [33]. Our results are not consistent with the mentioned study but are consistent with the results of another study [34]. Seed maturity significantly affected oleic acid, linoleic acid, and palmitic acid contents [35]. The increase in the oleic acid content with increasing seed maturity was often accompanied by a decrease in palmitic and linoleic acid [36], which was also observed in another study [34]. The variation in the oleic acid content within the same genotype suggested epistatic regulation or the contribution of other *FAD2* genes [37].

FAD2 is among the best-studied plant fatty acid desaturase gene families in terms of both molecular and biochemical investigations. Although only a single FAD2 was identified in *Arabidopsis*, FAD2 appears to exist as a complex gene family in most plant species [38,39]. There were seven different FAD2-encoding genes in peanut reported by previous researchers and newly cloned in this study [17,22,23]. Similar to other plant membrane desaturases, all peanut FAD2 polypeptides have three conserved histidine centers in hydrophilic regions [39]. These histidine boxes, such as HX(3-4)HH, HX(2-3)HH, and HX(2-3)HH, are located sequentially from the N-terminus to the C-terminus, constitute the catalytic center of the enzyme, and form ligands to a diiron-oxygen cluster in the catalytic site [40]. FAD2 lacks an ER retention signal peptide with the consensus motif (H/R/K)DEL at its C-terminus, although it is located in ER membranes. However, similar to other plant FAD2 proteins, AhFAD2-7, AhFAD2-8, and AhFAD2-9 were found to have an aromatic-amino-acid-enriched signal at the C-terminus of the proteins, which has been reported to be essential for maintaining localization of the enzymes in the ER [41].

Constitutive and seed-specific expression patterns have been observed for FAD2 genes in many studies [4,5]. Among the five *FAD2* genes detected in this study, *AhFAD2A/B* and *AhFAD2-3* were expressed mainly in seeds and flowers. *AhFAD2-7*, *AhFAD2-8*, and *AhFAD2-9* were expressed constitutively throughout the plant tissues [42]. Many studies have reported that *AhFAD2A/B* are major genes to control the linoleic acid content in peanut [17,22]. Compared with other *AhFAD2* genes, *AhFAD2A/B* were expressed at the highest level in developing seeds, which was similar to the expression patterns of *AhFAD2-1* genes in most plants, such as soybean and cotton [8,11]. The expression level of *AhFAD2A/B* was not correlated with the linoleic acid content of peanut seeds of the AABB genotype at different developmental stages, and the variation trend was the same in the seeds with the aabb genotype. Therefore, *AhFAD2A/B* may be not involved in maturity-related linoleic acid accumulation. The expression patterns of *AhFAD2-3* were similar to those of *AhFAD2A/B* in four genotypes. It is interesting to note that the expression of *AhFAD2-7* was the second highest in developing peanut seeds, while the expression of *AhFAD2-7* was higher in vegetative tissues. With the increase in seed maturity, the expression of *AhFAD2-7* decreased, which coincided with the changing trend of the linoleic acid content. Although likely contributing to the overall oleate desaturase activity in peanut vegetative tissues, AhFAD2-8 and AhFAD2-9 presumably play only a secondary role in linoleic acid production in seed oil. Furthermore, the expression levels of *AhFAD2-8* and *AhFAD2-9* increased with the degree of seed maturity, which was in contrast with the change in the linoleic acid content. In addition, *AhFAD2-8* and *AhFAD2-9* were preferentially expressed in flowers, similar to cotton *GhFAD2-1* and safflower *CtFAD2-10* [43,44]. Taken together, all the *AhFAD2* genes showed tissue preference expression to some extent. We found that the expression level of AhFAD2-7 is positively correlated to the linoleic acid concentration in peanut seeds, indicating that AhFAD2-7 may play a significant role in the accumulation and regulation of linoleic acid related to seed maturity.

## 5. Conclusions

In summary, our study suggested that the genotypes of *AhFAD2A/B* and seed maturity significantly affect the contents of oleic acid, linoleic acid, and palmitic acid. The oleic acid content increases with seed maturity, while the changes in linoleic acid and palmitic acid contents are opposite to that of the oleic acid content. Three new AhFAD2 genes (*AhFAD2-7*, *AhFAD2-8*, and *AhFAD2-9*) were identified. These *AhFAD2* genes show different expression patterns, although they display tissue preference. The expression of *AhFAD2-7* is positively correlated with the linoleic acid concentration in peanut seeds during maturation.

## Figures and Tables

**Figure 1 genes-13-02076-f001:**
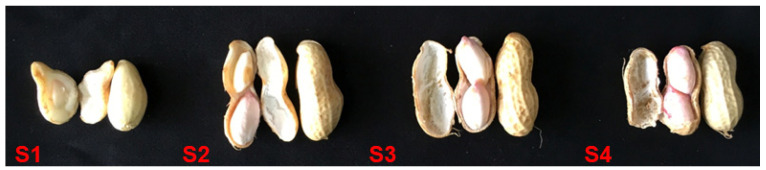
Four different developmental stages of peanut pods. S1: white and thick skin stage; S2: white and thin skin stage; S3: brown and thin skin stage; S4: black and thin skin stage.

**Figure 2 genes-13-02076-f002:**
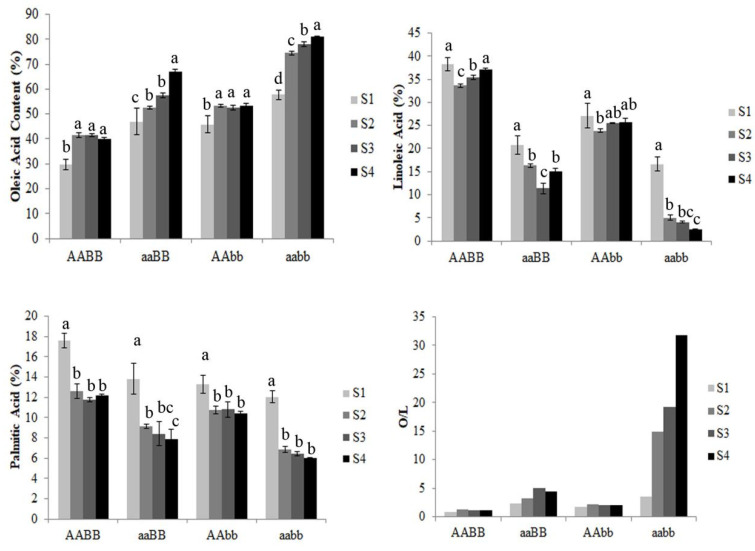
Fatty acid content and oleic-acid-to-linoleic-acid ratio of four genotypes AABB, aaBB, AAbb, aabb in different developmental stages of peanut seeds.

**Figure 3 genes-13-02076-f003:**
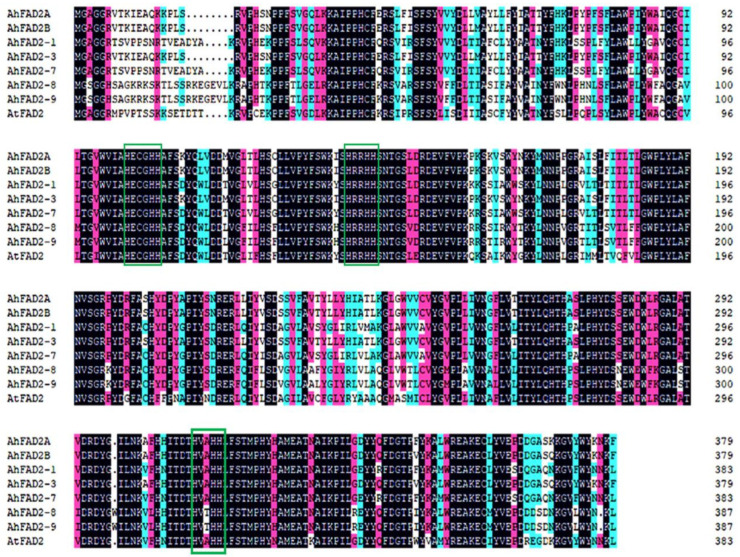
Alignment of FAD2 proteins from different plant species. Black shading indicates conserved amino acids. Green boxes represent H boxes.

**Figure 4 genes-13-02076-f004:**
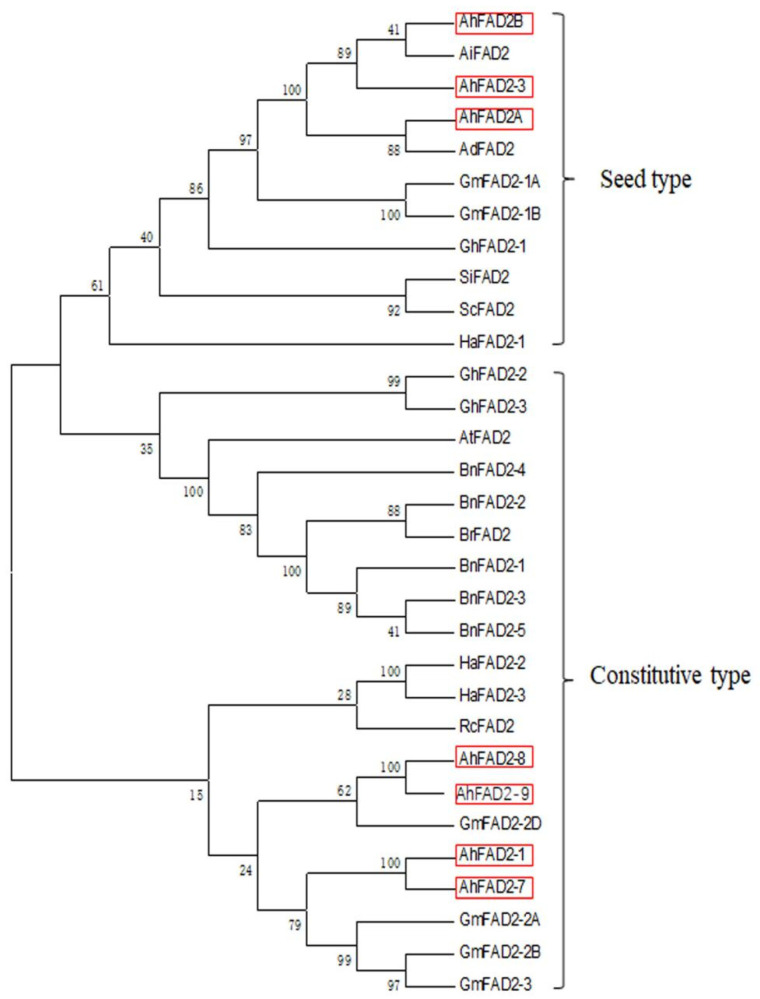
Phylogenetic tree analysis of plant microsomal oleate desaturases: *Arabidopsis thaliana* (AtFAD2, L26296), *Arachis duranensis* (AdFAD2, AF272951), *Arachis ipaensis* (AiFAD2, AF272952), *Brassica rapa* (BrFAD2, AJ459107), *Brassica napus* (BnFAD2-1, FJ907397; BnFAD2-2, FJ907398; BnFAD2-3, FJ907399; BnFAD2-4, FJ907401; BnFAD2-5, FJ907400), *Glycine max* (GmFAD2-1A, L43920; GmFAD2-1B, AB188251; GmFAD2-2A, XM_006605157; GmFAD2-2B, NM_001360081; GmFAD2-2D, XM_014762092; GmFAD2-3, DQ532371), *Gossypium hirsutum* (GhFAD2-1, X97016; GhFAD2-2, Y10112; GhFAD2-3, AF331163), *Helianthus annuus* (HaFAD2-1, AF251842; HaFAD2-2, AF251843; HaFAD2-3, AF251844), *Ricinus communis* (RcFAD2, MK033959), *Sesamum indicum* (SiFAD2, AF192486), and *Solanum commersonii* (ScFAD2, X92847).

**Figure 5 genes-13-02076-f005:**
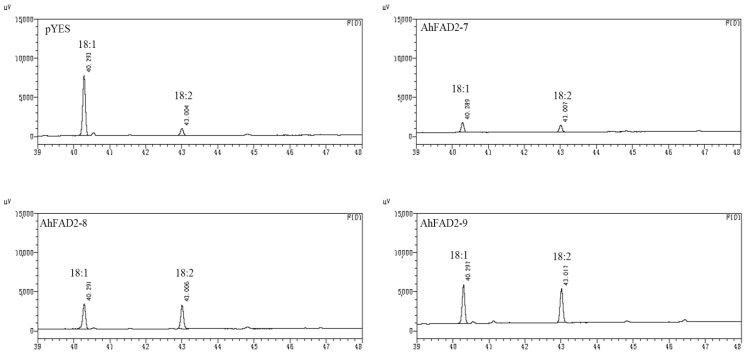
GC-MS of yeast cultures. pYES2 (control), pYFAD2-7 (pYES2 transformed with AhFAD2-7), pYFAD2-8 (pYES2 transformed with AhFAD2-8), and pYFAD2-9 (pYES2 transformed with AhFAD2-9). C18:1, oleic acid; C18:2, linoleic acid.

**Figure 6 genes-13-02076-f006:**
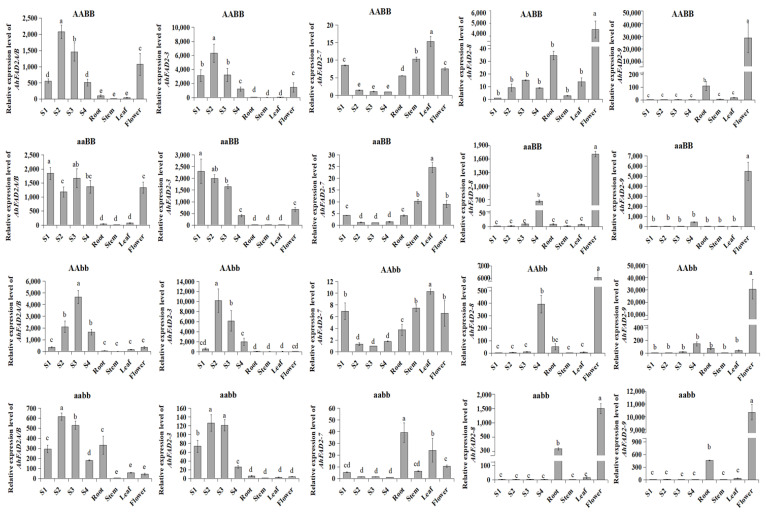
Expression analysis of five desaturase genes in different tissues of four genotypes using qRT-PCR. The bars are standard deviations (SDs) of three technical repeats. Different letters within a column indicate a significant difference at *p* < 0.05.

**Table 1 genes-13-02076-t001:** Fatty acid composition of transformed *S. cerevisiae* INVSc1.

Transformant	Oleic Acid (%)	Linoleic Acid (%)
pYES2	55.79	1.70
pYFAD2-7	47.22	10.44
pYFAD2-8	13.42	49.14
pYFAD2-9	12.87	57.03

**Table 2 genes-13-02076-t002:** Determination of fatty acids in transgenic Arabidopsis.

	Oleic Acid (%)	Linoleic Acid (%)
WT	11.63	31.01
*fad2*	57.28	2.56
AhFAD2-7	44.23	7.49
AhFAD2-8	51.45	3.82
AhFAD2-9	39.60	13.06

## Data Availability

Not applicable.

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
