# Peer review of "Insights into the Novel FAD2 Gene Regulating Oleic Acid Accumulation in Peanut Seeds with Different Maturity"

_genes, 2022, doi:10.3390/genes13112076_

Round 1
Reviewer 1 Report
Dear authors,
The paper is mostly fine, but my points are listed below.
English in the Introduction needs to be improved. Check plural/singular, tenses and the other grammar.
What is relationship between FAD2A-FAD2B and FAD2-x nomenclatures? Are FAD2A/B also any FAD2-x?
Abbreviations need to be listed.
ER Retrieval retrieval from ER or to ER? Please write explicitly.
AhFAD2-8A and AhFAD2-8B appear only once and need explanations.
Author Response
Dear Editor and Reviewer,
Thank you very much for the opportunity to revise our manuscript ( genes-1998918) entitled "Insights into the novel FAD2 gene regulating oleic acid accumulation in peanut seeds with different maturity". We value the comments received greatly, as they have pointed out important issues before further processing. We have now incorporated several modifications to the initial manuscript following your comments and suggestions. The modified and added texts in the revised manuscript were marked up using the “Track Changes” function. Following this letter, the reviewer’s comments are reproduced in black with our point-by-point responses in blue.
- English in the Introduction needs to be improved. Check plural/singular, tenses and the other grammar.
Answer: Thank you for your suggestions. We revised the Introduction in the revised version.
- What is relationship between FAD2A-FAD2B and FAD2-x nomenclatures? Are FAD2A/B also any FAD2-x?
Answer: FAD2A and FAD2B are members of FAD2 family. The two genes were isolated by Jung et al. in 2000 (Jung et al. 2000), and named as FAD2A and FAD2B, therefore, I didn’t change their names as FAD2-x. The reference is as below:
Jung, S.; Swift, D.; Sengoku, E.; Patel, M.; Teule, F.; Powell, G.; Moore, K.; Abbott, A. The high oleate trait in the cultivated peanut (Arachis hypogaea L.). Isolation and characterization of two genes encoding microsomal oleoyl-PC desaturases. Mol Gen Genet. 2000, 263, 796–805.
- Abbreviations need to be listed.
Answer: Thank you for your suggestions. Where the abbreviation first appeared in the text, we added its full name.
- ER Retrieval retrieval from ER or to ER? Please write explicitly.
Answer: Thank you for your suggestions. ER retrieval motif of FAD2 could maitain FAD2 protein locate to ER.
- AhFAD2-8A and AhFAD2-8B appear only once and need explanations.
Answer: Thank you for your suggestions. We missed the revision in Section 3.3. AhFAD2-8A and AhFAD2-8B were changed with AhFAD2-8 and AhFAD2-9, respectively.

Reviewer 2 Report
In this study, the author characterized three novel FAD2 genes in peanut. The result is solid and readable. Some comments are as follows:
1. Page 1, line2: In the title, the novel FAD2 gene will let reader to have the first impression that the authors studied only one gene. However, you actually characterized three FAD2 genes in peanuts. Maybe the title can be slightly modified.
2. Page 2, line 82: the authors explained O/L in page 4, lines 183-184. Please use full name for the first time.
3. Page 3, Section 2.2, lines 105-118: Please indicate the database accessed dates.
4. Page 3, line 121: Please provide the company name, city name, and country name of Aidlab RNA extract kit.
5. Page 3, line 121: Please add city name in between Takara and China..
6. Page 3, line 136: Who is the supplier of SYBR Green Master Mix? Also you should provide company name, city name, and country for the ABI 7500 real-time PCR system.
7. Page 4, line 143: 2 mL can be replaced by Two mL.
8. Page 4, line 145: For consistent with rifampin, Kan can be replaced by kanamycin.
9. Page 4, line 151: Use the full name, Saccharomyces cerevisiae, for the first time.
10. Page 7, line 256: linoleic acid should be capitalized.
11. Page 8, line 267: Please add a space in between ‘3.82%’ and ‘and’.
12. Page 9, Figure 6: the resolution and quality need to be improved. Try to enlarge the subfigures for entire page but not just half pahe.
13. Page 9, line 297: 4. Discussion.
14. Page 10, lines 349, 352, 353, and 361: Four LA are odd. I personally like the full name, linoleic acid, showing in line 337, 344, 346, etc.
15. Page 10, line 359: ‘these’ should be capitalized.
16. Page 11, line 403: unwanted space in this reference.
17. This is my personal opinion, and the authors can either ignore or accept. I think the four supplementary tables are important and be considered to place in the main text especially Table S3 and S4 are important results to support the potential functions of the gene products both in yeast and Arabidopsis. In the title of Table S3, S. cerevisiae cannot be omitted.
Author Response
Dear Editor and Reviewer,
Thank you very much for the opportunity to revise our manuscript ( genes-1998918) entitled "Insights into the novel FAD2 gene regulating oleic acid accumulation in peanut seeds with different maturity". We value the comments received greatly, as they have pointed out important issues before further processing. We have now incorporated several modifications to the initial manuscript following your comments and suggestions. The modified and added texts in the revised manuscript were marked up using the “Track Changes” function. Following this letter, the reviewer’s comments are reproduced in black with our point-by-point responses in blue.
- Page 1, line2: In the title, the novel FAD2 gene will let reader to have the first impression that the authors studied only one gene. However, you actually characterized three FAD2 genes in peanuts. Maybe the title can be slightly modified.
Answer: Thank you for your suggestions. Yes, we characterized three FAD2 genes including AhFAD2-7, AhFAD2-8 and AhFAD2-9 in peanuts, however, according to our results, only AhFAD2-7 gene regulated oleic acid accumulation in peanut seeds with different maturity.
- Page 2, line 82: the authors explained O/L in page 4, lines 183-184. Please use full name for the first time.
Answer: Thank you for your suggestions. We revised it in the revised version.
- Page 3, Section 2.2, lines 105-118: Please indicate the database accessed dates.
Answer: The database accessed date was July fifth, 2021.
- Page 3, line 121: Please provide the company name, city name, and country name of Aidlab RNA extract kit.
Answer: Thank you for your suggestions. We revised them in the revised version.
- Page 3, line 121: Please add city name in between Takara and China..
Answer: Thank you for your suggestions. We revised it in the revised version.
- Page 3, line 136: Who is the supplier of SYBR Green Master Mix? Also you should provide company name, city name, and country for the ABI 7500 real-time PCR system.
Answer: Thank you for your suggestions. We revised them in the revised version.
- Page 4, line 143: 2 mL can be replaced by Two mL.
Answer: Thank you for your suggestions. We revised it in the revised version.
- Page 4, line 145: For consistent with rifampin, Kan can be replaced by kanamycin.
Answer: Thank you for your suggestions. We revised it in the revised version.
- Page 4, line 151: Use the full name, Saccharomyces cerevisiae, for the first time.
Answer: Thank you for your suggestions. We revised it in the revised version.
- Page 7, line 256: linoleic acid should be capitalized.
Answer: Thank you for your suggestions. We revised it in the revised version.
- Page 8, line 267: Please add a space in between ‘3.82%’ and ‘and’.
Answer: Thank you for your suggestions. We revised it in the revised version.
- Page 9, Figure 6: the resolution and quality need to be improved. Try to enlarge the subfigures for entire page but not just half pahe.
Answer: We improved the resolution and quality of figure 6 in the revision.
- Page 9, line 297: 4. Discussion.
Answer: Thank you for your suggestions. We revised it in the revised version.
- Page 10, lines 349, 352, 353, and 361: Four LA are odd. I personally like the full name, linoleic acid, showing in line 337, 344, 346, etc.
Answer: Thank you for your suggestions. We replaced LA with linoleic acid in the revised version.
- Page 10, line 359: ‘these’ should be capitalized.
Answer: Thank you for your suggestions. We revised “These” in the revised version.
- Page 11, line 403: unwanted space in this reference.
Answer: Thank you for your suggestions. We revised it in the revised version.
- This is my personal opinion, and the authors can either ignore or accept. I think the four supplementary tables are important and be considered to place in the main text especially Table S3 and S4 are important results to support the potential functions of the gene products both in yeast and Arabidopsis. In the title of Table S3, S. cerevisiae cannot be omitted.
Answer: Thank you for your suggestions. We added “Table S3 and S4” in the main text and named them as Table 1 and Table 2 in the revised version, respectively.

Round 2
Reviewer 2 Report
The authors responded all my comments. To me, this manuscript is acceptable.